# OpenReview forum: "Adversarial Suffixes May Be Features Too!"
_ICLR.cc/2025/Conference — ICLR 2025 Conference Withdrawn Submission_

### Official Review · Reviewer_LWhv · 2024-11-01

**Soundness:** 2
**Presentation:** 3
**Contribution:** 2
**Rating:** 5
**Confidence:** 3

**Summary:**

The paper reports on a finding that suffixes
that encourage certain benign behavior, can also
cause a model to respond in unsafe ways.  It
also reports that fine-tuning a model on certain
benign datasets, can cause the fine-tuned model
to become less safe.

**Strengths:**

The paper describes some interesting phenomena
related to safety alignment, and reports on the
results of several experiments.  The results are
surprising.

These results have implication for safety
alignment and limitation of current alignment
methods.

**Weaknesses:**

I'm not sure how significant the results are.
I'm struggling a bit to synthesize what their
implications are and what I've learned.  This
might be my own shortcomings.

I find the framing challenging to understand.
I don't understand in what sense the suffix is
a "feature", or what it means to call a suffix
a "feature not a bug".

I find the use of suffixes that are embedding
vectors (instead of tokens) fairly unconvincing.

I am concerned the "LLM as judge" may be inaccurate
and may be classifying harmless responses as safety
violations.  For the 4 examples shown in this paper,
in my opinion half of them involve such erroneous
classifications.  This might throw off the paper's
results.

Detailed comments

Fig.1: Are these real responses, or did you make them up?
If real, have you copied them accurately?  e.g., "converving"
Rather than <suffix representing...>, can you show the actual
suffix?  What is the rightmost column representing, how
should I interpret it, and why is it there?

Fig.1: The caption claims that the middle column violates
safety alignment, but I don't see any violation of alignment.
There is no racist joke.  I don't think it violates safety
or any safety policy to respond in the way shown here.

Sec 2.1: Why is it reasonable to suffixes to be embedding
vectors that don't correspond to any input token?  GCG
generates adversarial suffixes that are input tokens.
You might be finding embedding vectors that don't correspond
to any input token, and if so, I don't see how that is
very relevant or provides strong enough evidence in
support of your hypothesis.

Regularization (2) does not suffice to ensure that the
embedding corresponds to any input token.  The paper
claims it "encourages the optimized suffix embeddings"
to correspond to valid token sequences (and "encouraging
the generation of valid token embeddings"), but that is
an overstatement; it doesn't do that.  It encourages
proximity to embeddings of valid tokens, which is not
the same thing.

Similarly, when you compute a suffix (embedding vectors)
using regularization (2), it is misleading to call it a
"token sequence" when it is actually an embedding vector
(that cannot be exactly produced by any sequence of
tokens) rather than a sequence of input tokens.

Sec 3.1: Is there any overlap between the set of 1000
benign prompts used for suffix generation and the 500
prompts used for evaluating transferability?

Sec 4.2: I'm not clear on what suffixes you used
for these experiments.  Are they suffixes generated
to produce poetry outputs?  to produce structured
outputs?  suffixes generated by GCG?  etc.?

I'm confused.  Don't you generate the suffix
to induce responses that are poems or lists?  It seems
weird that a suffix optimized to produce lists would
in practice produce BASIC programs (for example).  What
is going on here?

The name of the programming language is BASIC, not
"Basic" or "basic".  It is confusing to use "basic"
when you mean BASIC.

You say 3 suffixes trigger a distinct response style.
What is the denominator?  In other words, how many
suffixes did you try, to find 3 that have this
property?

Section 5.1: I don't understand how using Llama3guard
makes sense.  Llamaguard is not designed to respond to
arbitrary prompts.  Instead, it is designed as a
classifier, to classify input as toxic or not.

Fig.14,16: This does not violate OpenAI's policies
and is not a safety violation.

**Questions:**

Can you pick a random sample of responses judged
harmful, and manually rate whether they are harmful
(e.g., violate OpenAI policies), to validate the
accuracy of your judge?  I'm concerned it might
not be very accurate.

Can you randomly sample 10 responses judged harmful,
and show them to us, so we can form our own opinion?

Are you able to demonstrate similar results with
suffixes that are tokens, not embedding vectors?

---

### Official Review · Reviewer_XayL · 2024-11-03

**Soundness:** 2
**Presentation:** 3
**Contribution:** 2
**Rating:** 5
**Confidence:** 4

**Summary:**

This work attempts to empirically demonstrate some correlation between adversarial suffixes (generated by an automated jailbreak attack) and benign instructions (e.g., ones that instruct the model to respond in some specific format). The authors conduct three main sets of experiments to demonstrate this phenomenon from different perspectives.

**Strengths:**

1. I believe that the first experiment has the most surprising and scientifically interesting result. This suggests that adversarial suffixes do not simply work by directly “reverting” the alignment process but also by taking the model out of the usual distribution (I suspect that the model is not safety-tuned / aligned under a very broad set of instructions and prompts).
2. It is also interesting to see that different datasets, though all seemingly benign, can have a very different effect on the model on which they were fine-tuned (Experiment 3).
3. The experiments are mostly well-designed and do support the main hypothesis of the paper.
4. I’m quite surprised that the nearest token project works at all in Algorithm 1. From personal experience and reading a few related papers [[1](https://arxiv.org/abs/2104.13733), [2](https://arxiv.org/abs/2405.09113), [3](https://arxiv.org/abs/2410.04234)] in this domain, I believe that naively projecting embeddings to tokens has not found much success. Although this is slightly out of scope for this paper, it would be interesting to deep dive into why this algorithm works and the others have failed.

**Weaknesses:**

1. **Additional experiments.** I believe that this paper contains interesting observations, but I would really want to see more extensive experiments and a deeper dive into the underlying reasons.
    1. I hope to see more response formats/styles in Experiment 1. What about response as a song, with some persona, in a different languages, etc.?
    2. Is there any correlation between these formats and the datasets used for aligning the model? Perhaps, these formats are never seen by the model during safety tuning or RLHF, which makes them OOD and can easily avoid the refusal?
    3. In Experiment 1, Llama-2 is the only model that underwent safety tuning (I believe that Vicuna and Mistral are only instruction-tuned). I’d like to see whether this observation applies broadly to aligned models.
    4. Experiment 1 only covers the embedding attack. I’m curious whether the result also extends to the token-level attack (hard prompt)?
2. **Other adversarial suffix generation algorithms**
    1. I don’t think I understand the point of Algorithm 1. There are many existing prompt optimization methods for both hard and soft prompts. So why do the authors also propose a new algorithm?
    2. I’m also wondering whether the adversarial suffixes generated from different algorithms also exhibit a similar phenomenon. Is there anything special with Algorithm 1? (This is related to Question 5 below).
3. **Novelty**
    1. Experiment 3 is not entirely novel as it is a well-known phenomenon today.
    2. If we know that fine-tuning can undo the alignment of LLMs, it is perhaps not very surprising that prompt tuning could also undo the alignment as well. Experiment 1 is essentially equivalent to soft prompt tuning.
4. **PCC metric.** Since X, Y are vectors, cov(X, Y) should be a covariance matrix? Based on the caption of Figure 2, it seems like each dot is one PCC value computed on a pair of prompt and suffix. If my understanding is correct, this is treating each dimension of the embedding as a "sample" which seems wrong to me. The right interpretation would be to compute PCC of one embedding dimension across all the samples (pairs of prompt and suffix). That said, it seems difficult to interpret this number so my suggestion is to just compute a distance (Euclidean or cosine) between the two vectors and just average over all the pairs/samples.

### Nitpicks

1. Figure 1: This example is not that clear to me.
2. Section 3.1 (Dataset Construction): It is unclear to me how this experiment setup has to do with "generate suffixes that capture benign features." I'd suggest at least describing the purpose of the two constructed datasets, what they represent, what the desired outcomes are, etc.
3. Section 3.1 and the term "feature extraction": This terminology is also quite misleading to me; When seeing the term, I expect that we will take some intermediate representation of a neural
network given some inputs. However, I think "feature extraction" here refers to the conceptual "feature" of the suffix (like "cat features", "dog features" in the computer vision domain) -- There's no "extraction" here I believe.

**Questions:**

1. L243 (”We apply the method presented in Section 2 to generate suffix embeddings…”): What is the target response y? I assume that it is the one generated by Llama2-7B-chat-hf above. I just want to confirm that this is correct.
2. L248 (”For each dataset, we generate one suffix in the form of embedding…”): Is this optimized with all samples at once (like universal attack) or just take the best embedding out of N where N is the dataset size? How is the "best performance" exactly determined here?
3. L312 (”We first apply our method presented in Section 2 to construct multiple suffixes…”): How many suffixes are generated using Algorithm 1? Why are the numbers different between GCG (1k) and AmpleGCG (5k)?
4. L323: GCG generates only one for each prompt-response pair? Not universal, i.e., optimizing for multiple pairs at the same time?
5. Section 4.2: Does suffixes from GCG and AmpleGCG also exhibit any response style/format pattern as observed in suffixes from Algorithm 1? If not, what makes it different?
6. L430 (”three constructed based on the embedding universal adversarial suffixes generated in Experiment 2.”): What are the reasons for using these datasets from Experiment 2? Is the intention to say that some response formats/styles are better correlated with harmfulness or are just better at undoing the alignment?

---

### Official Review · Reviewer_37b8 · 2024-11-04

**Soundness:** 1
**Presentation:** 2
**Contribution:** 1
**Rating:** 3
**Confidence:** 5

**Summary:**

The paper proposes a modification to the usual adversarial suffix generation methods in the form of a regularizing term loss in the loss that increases based on the distance between the embedding of the adversarial suffix and the k nearest embedded datapoints. They find that if they fit a suffix from a dataset of, ex, poems, then conditioning the model on this suffix makes the model both output something poem-formatted (thus satisfying the criterion that the suffix has a "benign" property) but also that it induces a harmful output (so it has a "malign" property).

**Strengths:**

See below

**Weaknesses:**

I learned nothing new after reading the paper so I consider the contribution very weak and the methodology unsound. The writing and lack of reference to prior work indicates to me that the authors are not familiar with the latest papers in the field. This is to be expected because of the overwhelming quantity of jailbreak papers accepted at conferences. But it does mean that this work will not have much impact.

The paper evaluates the suffix based on a) whether it produces the desired style and b) whether it is harmful. However, it's well known that roleplay-style jailbreaks will already work to induce harmful outputs. This is because the model is not used to knowing that it should map {harmful question} -> {refusal} in the role-played setting. So I don't think the paper has any contribution beyond that of the many papers that analyze the impact of roleplaying on inducing harmful outputs. At least if the authors could have showed that this suffix is somehow "better" than a roleplay prompt, that could have been something, but I didn't see that anywhere.

I don't like the evaluation template that's used in Appendix A.1 and no comparison is done between this newly introduced evaluator and previously proposed evaluators or human judges.

Some of the evaluation is done on models like Vicuna and Mistral which are not aligned and can always be trivially jailbroken.

The analysis that employs the Pearson correlation coefficient is incorrectly used to establish causation in the paragraph starting on Line 296. When introducing this, the authors note that they are computing correlations between the hidden state representations. The hidden state representation is the model's representation of the first token that will be decoded conditioned on the prompt, where the prompt consists of either a prefix (H_p), a suffix appended to a prefix (H_p+s), or just the suffix (H_s). And they state if the hidden state representation of the prefix+suffix (H_p+s) is more correlated with the representation of just the suffix (H_s) than the representation of just the prefix (H_s), then this means the generated adversarial suffix had more of a role in shaping the model's output than the prefix, where the prefix is the harmful instruction. But I don't agree with this. After all, the hidden state representation that's being captured here is just the first token that's being decoded. That first token might be a newline character if the suffix was fitted on Structure. But this doesn't tell us that the entire model's output is mostly dependent on the suffix. We'd need to do this analysis based on the entire model output.

The evaluation of the effectiveness of the adversarial suffix uses GCG and a variant of GCG. A number of suffixes are generated and then the "best" one is chosen. But it's not clear that there is a fair comparison. To be clear I don't doubt that this method is better than GCG, which is the literal first UAR ever proposed. I just want to see some evidence that the comparison is fair.

The authors note that the more important insight from the evaluation in their view is that the method is able to generate very successful embedding suffixes. This is pretty much trivial since the embedding space is continuous and the entire difficulty of the suffix optimization is that it's discrete, but this also enables the suffix to be used on APIs where embeddings can't be provided directly to the model. I don't really follow the conclusion of the paragraph starting on Line 343 (first paragraph in Section 4) where they state that their goal is to show that the embedding suffixes generated by the attack are effective in inducing harmful outputs, but it's not to conduct an adversarial suffix attack.

The section on finetuning LLMs on benign datasets compromising safety is well established by Qi et al 2023 and He et al 2024 [1]. I found nothing new by reading this section. The authors should include a comparison to these works and probably there are more that cite them; there have been many many many papers written on the compromise of safety due to finetuning such as Qi et al 2024 [2].

There are some minor typos (Line 339) (Lines 859-862).

Putting screenshots of the OpenAI dashboard in the paper is pretty off-putting.

If I look at examples from the datasets I'm not convinced that the optimized suffix isn't learning something else. Ex, consider the example on line 986. It could just be that the suffix has distilled that the response should be very long.

Attack papers should generally provide some commentary on defenses, and I saw no such commentary.

[1] https://arxiv.org/abs/2404.01099 (COLM 2024)
[2] https://arxiv.org/abs/2406.05946

**Questions:**

Please provide the corrected code link.

Did you evaluate the marginal benefit of the suffix over just instructing the model to respond in poetic verse?

---

### Official Review · Reviewer_N4Dm · 2024-11-05

**Soundness:** 3
**Presentation:** 2
**Contribution:** 2
**Rating:** 5
**Confidence:** 3

**Summary:**

The paper aims to draw connections to the famous Ilyas et al. paper on adversarial examples being the result of unusual, non-robust features in the image domain. The authors aim to do so by running experiments that PGD-style optimize prompts that aim to produce a specific pattern, and then viewing transferability of these patterns between benign prompts (where they produce the pattern and a benign response) and harmful prompts (where they produce the pattern and a jailbreak response). So the feature is essentially something that elicits the pattern.

**Strengths:**

- The choice of problem itself is interesting, there is little effort on any interpretability of adversarial suffixes in the literature.
- Extensive experiments to tackle each section. I don't think the experimental setups themselves have issues (although you could use more SoTA models)

**Weaknesses:**

- Maybe say soft prompt optimization when describing your embedding optimization strategy, that's the common term I believe.
- Your loss in Equation 3 may be related to a sparsity constraint from prior work. Take a look: https://arxiv.org/pdf/2405.09113
- The correlation analysis is a nice idea. But there may be problems - who is to say that PCC of 0 means little to no impact? It says there is no linear correlation, there may still be correlation. Any reason why you don't use existing token attribution techniques instead - I'm sure there must be some?
- Section 3.1 - This is rather poorly written, and took me several reads to understand. Correct me if I am wrong - you are constructing synthetic responses that enumerate/speak in poems to give an answer, and then optimize a suffix against some victim model so that it responds in this enumerated/poetic fashion. The reason it reads poorly is because its not clear what the target string is as per your soft prompt optimization strategy. You should probably explicitly state that you are optimizing the suffix to produce this unusual output, and that by default the victim model doesn't respond in this fashion. Would recommend being clear about this in the beginning of the section.
- Again Section 3.1 - In general, I am not sure why it is surprising that suffixes for benign prompts transfer to other prompts unless I see an example of the suffix. The prompt optimization objective seeks to maximize probability of specific token(s) in the output distribution, it could very well be that the suffix (in the token space) says something like "noBeGIN w#(with1234" - this is likely to cause an independence from the original prompt. I have seen suffixes like this when running GCG.
- Section 4.2 - Again, I am having a hard time understanding why this feature is interesting when there is (to me) a high likelihood of the story telling/repeat response/basic program directive being present in the tokens itself (when considering the suffix projected to the token space). In this case, the feature is simply some suggestive input-level tokens, no? It is really only insightful if the tokens don't obviously suggest these directives on their own. If it helps, the analogy in my head is, say you have a a binary dog vs. cat image classifier - if an adversarial perturbation applied to a dog image (to make it look like a cat) already resembles the cat, then why is it meaningful or interesting to study? There needs to be some way of characterizing the actual content of the suffixes.
- Also "wearisomely" is a typo.

**Questions:**

- Please address my 3 comments on the PCC, section 3.1, and 4.2. Perhaps I am misunderstanding something. I think the work has potential.

---

### Note · Authors · 2024-11-29

I have read and agree with the venue's withdrawal policy on behalf of myself and my co-authors.